# A Y-chromosome shredding gene drive for controlling pest vertebrate populations

Thomas AA Prowse[1†*], Fatwa Adikusuma[2,3†], Phillip Cassey[4,5], Paul Thomas[2,3*], Joshua V Ross[1]

[1]School of Mathematical Sciences, The University of Adelaide, Adelaide, Australia; [2]School of Medicine, The University of Adelaide, Adelaide, Australia; [3]South Australian Health and Medical Research Institute, Adelaide, Australia; [4]The Centre for Applied Conservation Science, The University of Adelaide, Adelaide, Australia; [5]School of Biological Sciences, The University of Adelaide, Adelaide, Australia

**Abstract** Self-replicating gene drives that modify sex ratios or infer a fitness cost could be used to control populations of invasive alien species. The targeted deletion of Y sex chromosomes using CRISPR technology offers a new approach for sex bias that could be incorporated within gene-drive designs. We introduce a novel gene-drive strategy termed Y-CHromosome deletion using Orthogonal Programmable Endonucleases (Y-CHOPE), incorporating a programmable endonuclease that 'shreds' the Y chromosome, thereby converting XY males into fertile XO females. Firstly, we demonstrate that the CRISPR/Cas12a system can eliminate the Y chromosome in embryonic stem cells with high efficiency (*c*. 90%). Next, using stochastic, individual-based models of a pest mouse population, we show that a Y-shredding drive that progressively depletes the pool of XY males could effect population eradication through mate limitation. Our molecular and modeling data suggest that a Y-CHOPE gene drive could be a viable tool for vertebrate pest control.
DOI: https://doi.org/10.7554/eLife.41873.001

**\*For correspondence:**
thomas.prowse@adelaide.edu.au
(TAAP);
paul.thomas@adelaide.edu.au (PT)

[†]These authors contributed equally to this work

**Competing interests:** The authors declare that no competing interests exist.

## Introduction

The engineering of 'selfish' gene drives, with biased inheritance (*Burt, 2003*), has become increasingly feasible since the advent of the CRISPR/Cas9 genome-editing system (*Esvelt et al., 2014*; *Gantz and Bier, 2015*; *Champer et al., 2016*; *Galizi et al., 2016*; *Hammond et al., 2016*; *Kyrou et al., 2018*). By spreading genetic elements through the genomes of wild populations, this technology could be used to address a range of environmental problems, including the control of invasive (or overabundant) sexually reproducing animal populations (*Prowse et al., 2017*; *McFarlane et al., 2018*). CRISPR/Cas9 gene drives are single-copy transgenic elements that encode the Cas9 endonuclease and a site-specific guide RNA (gRNA) that together act to cleave a matching target site on the homologous chromosome. Repair of the DNA break via homologous recombination (HR) results in replication (or 'homing') of the gene-drive construct (*Burt, 2003*; *Sinkins and Gould, 2006*). If the homing mechanism is triggered in the germline (*Bruck, 1957*; *Burt, 2003*), recessive traits that negatively impact the fitness of homozygous individuals should still spread though populations, and eventually lead to population suppression or eradication (*Deredec et al., 2008*; *Kyrou et al., 2018*).

Invasive alien species (IAS) threaten the conservation of biological diversity (*Bellard et al., 2016*; *Early et al., 2016*), impact human health (*Mazza et al., 2014*), and have severe economic consequences around the globe (*Pimentel et al., 2005*; *Hoffmann and Broadhurst, 2016*). In particular,

the human-mediated transport of vertebrate pests has contributed substantially to species' extinctions (*Woinarski et al., 2015*; *Spatz et al., 2017*; *Graham et al., 2018*) and loss of agricultural production (*Gong et al., 2009*). Consequently, there is considerable interest in the potential of gene-drive technology as a novel tool for controlling vertebrate pest populations (*Johnson et al., 2016*; *Prowse et al., 2017*; *Moro et al., 2018*), which could be used in combination with traditional control methods. Given the considerable negative impacts of rodent pests on island biodiversity (*Spatz et al., 2017*), and the risks associated with the unintended dispersal of gene-drive carriers beyond target populations (*Webber et al., 2015*), the eradication of invasive rodent populations on islands is a powerful model system for testing the efficacy of gene-drive technology (*Prowse et al., 2017*).

At their simplest, gene drives for population control could be designed to spread a recessive mutation that causes infertility or lethality (*Burt, 2003*; *Deredec et al., 2008*; *Gantz et al., 2015*; *Hammond et al., 2016*; *Kyrou et al., 2018*). To achieve this, the gene-drive cassette is positioned within the genome to inactivate a haplosufficient gene required for fertility or embryonic development. Assuming homing occurs in the germline, heterozygotes are unaffected and spread the drive through sexual reproduction, whereas the sterility or embryonic non-viability of homozygotes results in population suppression. *In silico* modeling has confirmed this approach could be viable for a number of insect species (*Burt, 2003*; *Deredec et al., 2008*; *Unckless et al., 2015*), and also small populations of invasive vertebrates (i.e. mice, rats and rabbits) on islands (*Prowse et al., 2017*; *Wilkins et al., 2018*).

An alternative strategy is to design a gene drive that achieves population suppression by biasing offspring sex ratios so that one sex become limiting (*Windbichler et al., 2008*; *Galizi et al., 2014*; *Galizi et al., 2016*; *Kyrou et al., 2018*). One approach to sex-ratio distortion is to introduce genetic 'cargo' which, when expressed, causes a phenotypic change during offspring development. In mice, for example, presence of the Y-linked *Sry* gene causes XX females to develop as sterile males (*Koopman et al., 1991*). Biased inheritance of an autosomally integrated *Sry* construct could be achieved by linkage to a naturally occurring selfish element (the t-haplotype; *Campbell et al., 2015*) or a synthetic gene-drive (*Prowse et al., 2017*). Since sterile heterozygous females cannot spread the construct, however, population suppression using this strategy requires the regular release of gene-drive carriers (*Backus and Gross, 2016*; *Prowse et al., 2017*). This drawback could potentially be overcome by incorporating the spermatogenesis genes required to convert females into fertile males; modeling indicates such a drive could achieve population eradication by reducing the availability of female breeding stock, but its spread would rely upon normal spermatogenesis in sex-reversed XX males (*Prowse et al., 2017*). Notably, *Kyrou et al. (2018)* recently demonstrated a simpler sex-distorting approach by engineering a CRISPR-based gene drive targeting the female version of the gene *doublesex* in the mosquito *Anopheles gambiae*. When introduced to caged mosquito populations at a frequency of 12.5%, the drive spread rapidly, converting homozygous females into sterile individuals with an intersex phenotype, and leading to total population collapse.

The targeted deletion of allosomes (X or Y sex chromosomes) offers another approach to sex reversal that could potentially be incorporated within gene-drive designs. To date, most research has focused on male-biasing X-shredding strategies, because in many species the abundance of mature females is a primary determinant of the population growth rate. In *Anopheles gambiae*, for example, a single-copy autosomal integration of the endonuclease I-PpoI was sufficient to shred the paternal X chromosome during meiosis, resulting in fertile males that produced >95% male offspring (*Galizi et al., 2014*). This approach effectively suppressed small caged populations of mosquitoes under a multiple-release strategy (*Galizi et al., 2014*). In the same species, a CRISPR-Cas9 system targeting an X-linked rDNA sequence achieved a male bias of between 86% and 95% (*Galizi et al., 2016*). X-shredding DNA cassettes could be spread by integration within the Y chromosome if X-shredding activity could be limited to the meiotic period (i.e. the Y-drive; *Hamilton, 1967*, *Deredec et al., 2011*, *Galizi et al., 2014*).

Recently, elimination of the murine *Y* chromosome in cultured cells and embryos has been achieved using a CRISPR/Cas9 system adapted from *Streptococcus pyogenes (Sp)* incorporating gRNA(s) that target multiple chromosomal locations simultaneously (*Adikusuma et al., 2017*; *Zuo et al., 2017*). The efficiency of Y chromosome shredding was high (up to 80%) and did not appear to negatively impact cell or embryo viability. Y-chromosome deletion in mice converts XY

males into XO females, which are not only viable but are also known to be fertile in the laboratory (*Kaufman, 1972*; *Probst et al., 2008*) and in the wild (*Searle and Jones, 2002*).

Here, we propose a strategy combining CRISPR-based gene drive and Y-chromosome deletion for generating female-biased sex ratios, which could be useful for population control. In this strategy, the CRISPR gene-drive cassette, which must be expressed strictly in the germ cells, also carries cargo that constitutively expresses the shredding machinery required to eliminate the Y chromosome at the zygote stage, a strategy we term Y-CHromosome deletion using Orthogonal Programmable Endonucleases (Y-CHOPE) (*Figure 1*). The spread of a Y-shredding drive would bias offspring sex ratios towards females through the production of XX- and XO-genotype offspring, and population control should be achieved once males become limiting (*Figure 1*). Since the homing and the Y-shredding processes occur at different times and target different sequences, a Y-CHOPE strategy would require two different CRISPR endonucleases. However, to date, only one programmable endonuclease (SpCas9) has been used for gene-drive homing and Y-shredding. Therefore, in this study, we also tested whether another commonly used CRISPR system, CRISPR/Cas12a (also known as Cpf1), could act as an efficient Y-shredder platform, thereby making a Y-CHOPE drive more feasible.

With invasive mice on islands as our motivating case study, we used *in silico* simulations to test the performance of two possible designs of a Y-chromosome shredding gene drive for vertebrate pest control. First, we assumed that the gene drive was placed within a 'safe harbour' non-coding region (hereafter referred to as 'non-coding' placement). However, it is known that such strategy is susceptible to the evolution of resistant alleles which can arise due to target sequence mutations when homing fails (*Prowse et al., 2017*; *Champer et al., 2018*). Therefore, we also tested a second design which positions the drive within a haploinsufficient developmental gene (hereafter placement

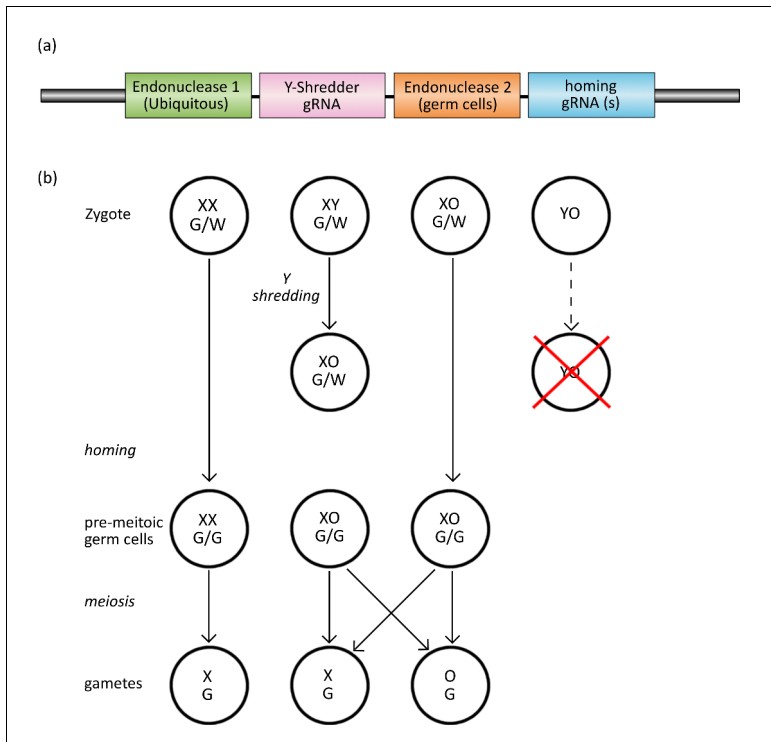

**Figure 1.** The Y-CHOPE drive. (a) The proposed Y-shredder gene drive construct. Y-shredder activity is mediated by a ubiquitously expressed programmable endonuclease (Endonuclease 1; e.g. Cas12a) and a gRNA targeting a Y chromosome repeat sequence. Homing activity is controlled by an orthogonal endonuclease (Endonuclease 2; e.g. Cas9) with restricted expression in the premeiotic germ cells, in combination with one or more gRNAs targeting the integration site of the transgene. (b) Chart of zygotic development depicting timing of Y-shredding, homing and meiosis. YO embryos fail to develop as indicated by the red cross.
DOI: https://doi.org/10.7554/eLife.41873.002

within a 'coding' region), which should ensure that the majority of deleterious mutations cause embryonic lethality of offspring that inherit them (*Esvelt et al., 2014*). Using stochastic, individual-based models that couple species' demography and genetic processes, we tested the ability of these designs to cause eradication, and explored how the population outcomes expected are influenced by the shredding efficiency of the drive, the fertility of XO females, the mating system of the target species, and the evolution of resistant alleles. We then used global sensitivity analysis to identify the simulation parameters with the greatest influence on the population outcomes expected, and used this information to comment on the technical feasibility the Y-shredding gene drive approach.

## Results

### Empirical Y-shredding efficiency

To investigate Cas12a-mediated Y chromosome deletion, we identified Cas12a gRNAs that target repeat sequences in the Y-centromere. Three potential centromere shredder gRNAs were tested: Centro 37X-A, Centro 37X-B and Centro 59X, each of which specifically targeted the Y-centromere at 37, 37 and 59 sites, respectively. Expression of Cas12a (from *Lachnospiraceae* bacterium) and each of the gRNAs in XY mouse embryonic stem cells resulted in deletion of the Y-chromosome with high efficiency compared to cells transfected with empty vector (EV) (*Figure 2*). In particular, the gRNA Centro 37X-A eliminated nearly 90% of the Y-chromosomes in the cell population based on

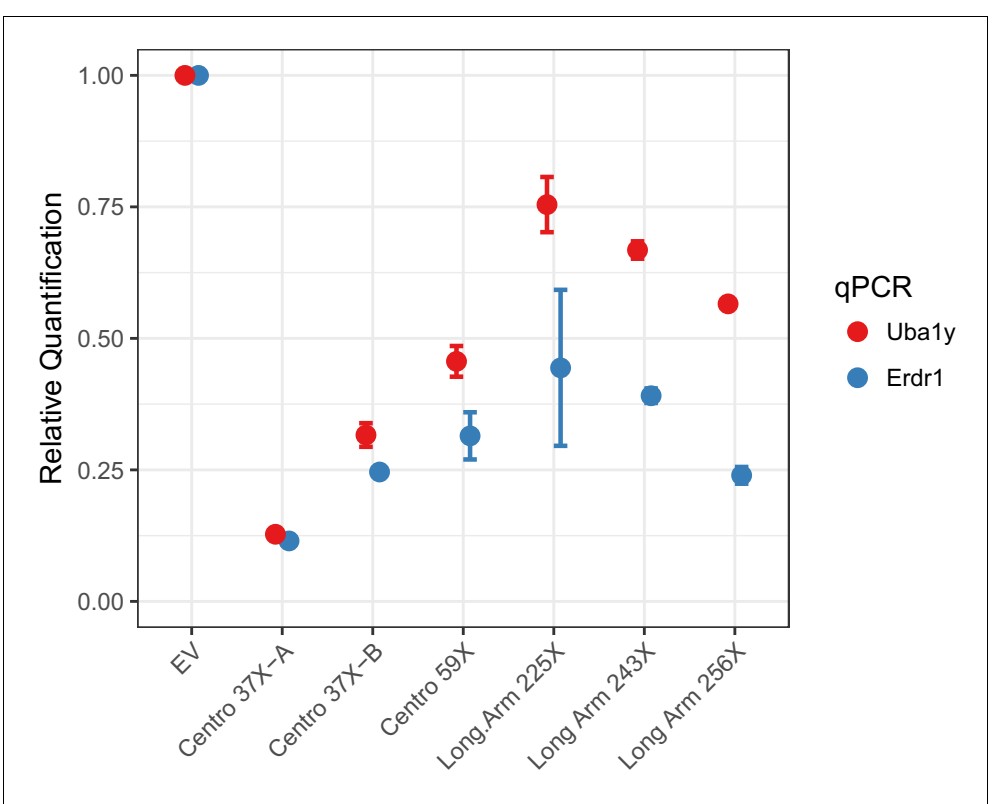

**Figure 2.** Empirical performance of the Y-shredding technology, using qPCR of genomic DNA to quantify Y-chromosome dosage. Controls were transfected with empty vector (EV). *Sox1* qPCR was used as the internal reference control. Data are presented as mean ±95% confidence intervals from n ≥ 3 biological replicates.
DOI: https://doi.org/10.7554/eLife.41873.004
The following source data is available for figure 2:

**Source data 1.** Empirical performance of the Y-shredding technology.
DOI: https://doi.org/10.7554/eLife.41873.003

qPCR analyses of sequences at the termini of short and long arms (*Figure 2*). This efficiency is slightly higher than our previously published Y chromosome deletion experiments using SpCas9 (~80% efficiency; *Adikusuma et al., 2017*).

We also assessed whether targeting long arm repeats using Cas12a could effect Y-chromosome elimination. Three Cas12a gRNAs specifically targeting the Y long arm 225, 243 and 256 times were assessed for Y-shredding as described above. All these gRNAs showed shredding activity, although none exceeded the efficiency of gRNA Centro 37X-A (*Figure 2*). We therefore conclude that centromere shredding using Cas12a provides an efficient approach for Y-chromosome deletion.

### *In silico* experiments

Individual-based simulations of the Y-chromosome shredding gene drive demonstrate that this strategy is capable of eradicating a pest vertebrate population (*Figure 3*). Using our baseline parameterization, eradication of a population of 10,000 mice was guaranteed when we assumed a perfect Y-shredding efficiency and 100% homologous recombination following Cas9 activity so that resistant alleles could not evolve (*Figure 3a(i)*). Under these conditions, the gene drive spread rapidly through the population, converting XY males to XO females until males became limiting and the population began to decline (*Figure 3*). However, the observed population trajectories changed as a function of the mating system simulated. Assuming short-term monogamy ($F_{max} = 1$), male limitation occurred rapidly and the population size declined smoothly from carrying capacity to extinction (*Figure 2b(i)*). In contrast, as higher levels of polygyny were simulated by increasing the maximum number of mates allowed per male, the sex-biasing effect of the drive increased the reproductive potential of the population initially (*Figure 2c(i)*). This caused a short period of population growth, until the proportion of females breeding was reduced due to male limitation, after which populations declined to extinction.

The simulated probability of eradication decreased as the Y-shredding efficiency was reduced, but there was a clear interaction with the level of polygyny assumed. Assuming $P_Y = 0.75$, for example, extinction was still guaranteed for a population exhibiting short-term monogamy (*Figure 3a(ii)*). Under polygyny breeding ($F_{max} = 3$), however, introduction of the gene drive produced population growth followed by suppression to a new stable state below carrying capacity. When $P_Y$ was reduced still further to 0.5, permanent population suppression (but not eradication) typically resulted under short-term monogamous mating (*Figure 3b(iii)*). However, population growth to a new carrying capacity (above the initial carrying capacity) was observed under polygynous breeding because the sex-ratio bias toward females increased reproductive potential without causing male limitation (*Figure 3c(iii)*). In these cases, the failure of the simulated eradication attempt under low Y-shredding efficiencies was not due to a lack of spread. In fact, the drive spread rapidly to fixation when non-homologous end-joining (NHEJ) was precluded ($P_N = 0.1$), but the sex-ratio distortion produced was insufficient to cause males to become limiting (*Figure 4*).

Placement of the gene drive within a non-coding region failed to produce simulated eradications when DNA mutations produced due to NHEJ was modeled ($P_N = 0.1$), even when perfect Y-shredding efficiency was assumed (*Figure 5a,b*). Under these conditions, the initial spread of the gene was reversed following the evolution of resistant alleles that could never acquire the drive. In such cases, gene-drive-positive alleles were purged from the population due to selection against sub-fertile XO females. However, the impact of NHEJ-mediated repair during homing could be negated by placing the gene drive within a haploinsufficient gene, such that any mutation arising through NHEJ causes a failure early in development ($P_{nf} = 1$). Using this gene-drive design, population eradication could be achieved with a high $P_N$ of 0.1 (*Figure 5c,d*).

The Y-shredding efficiency required to guarantee extinction increased with the level of polygyny assumed for mice. Assuming short-term monogamy, for example, Y-shredding efficiencies exceeding 0.67 could guarantee extinction (*Figure 6a*). In contrast, allowing up to 3 and 5 mates per male increases the Y-shredding efficiency required for certain eradication to 0.87 and 0.92, respectively (*Figure 6a*). The Y-linked X-shredding drive is not influenced by polygyny, because females become the limiting sex. However, our comparative simulations indicated that the suppression achieved by this strategy could be inferior to a Y-shredder when: (1) the breeding system of the target species approached short-term monogamy or (2) the probability of inheriting a Y chromosome from a germline YO male was similar to Mendelian rates (*Figure 6b*).

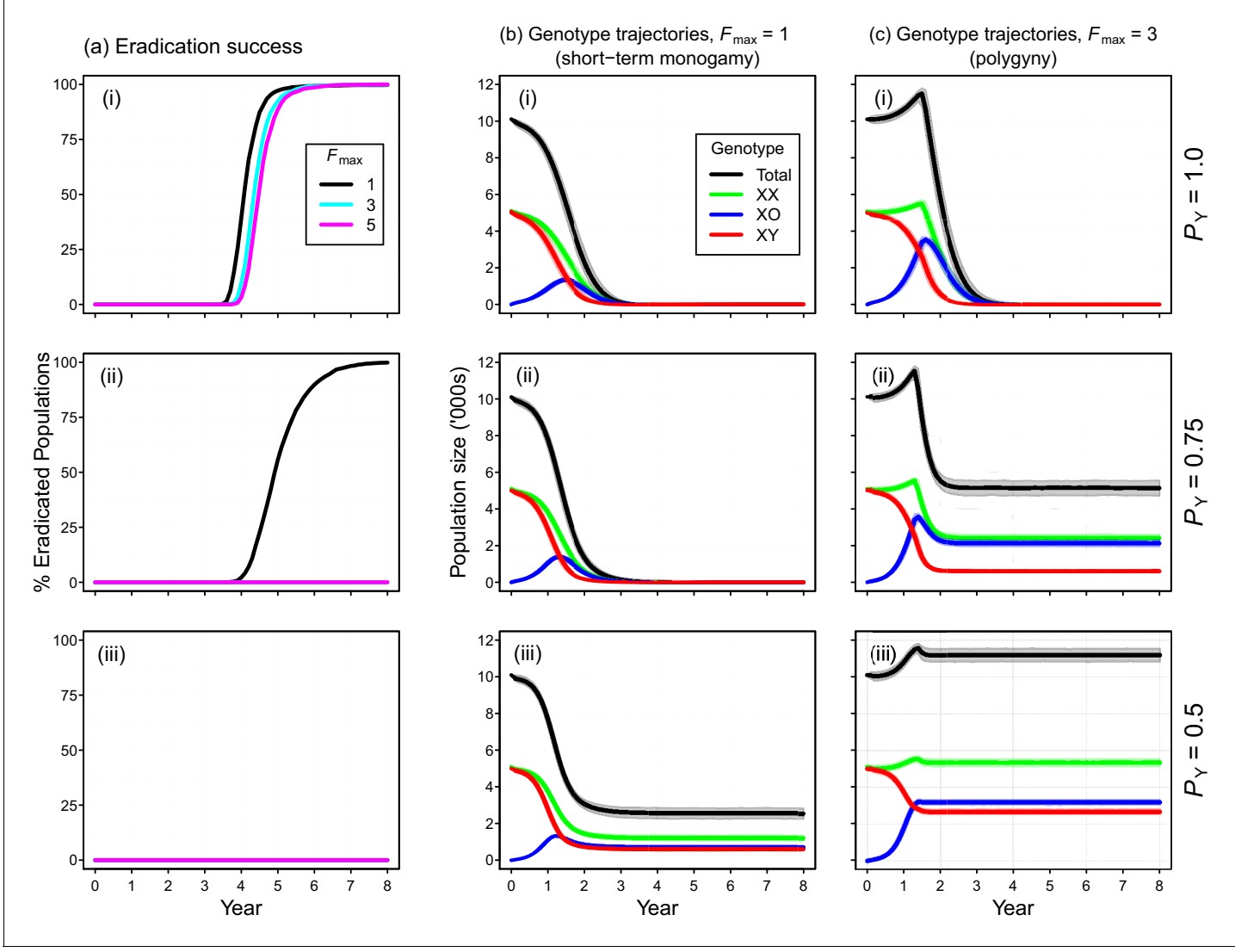

**Figure 3.** Performance of the Y-chromosome shredding gene drive for different Y-shredding efficiencies ($P_Y$), assuming resistant alleles cannot evolve. (a) The percentage of eradicated mouse populations over time, and different levels of polygyny ($F_{max}$). (b) Total population and allosome genotype trajectories (mean ±95% confidence intervals), assuming a short-term monogamous mating system for mice ($F_{max} = 1$). (c) As for (b), but assuming a polygynous mating system ($F_{max} = 3$). All results were based upon 1000 iterations per scenario, and assumed an initial population size of 10,000 mice that was inoculated with 100 gene-drive carrying XX females.

DOI: https://doi.org/10.7554/eLife.41873.005

Global sensitivity analysis for the Y-CHOPE strategy, using Latin hypercube sampling of the parameter space, demonstrated that the mating system of the target species ($F_{max}$) and the Y-shredding efficiency ($P_Y$) were the primary determinants of simulation outcomes, with relative-influence scores from the boosted regression tree BRT emulator of 29% and 28%, respectively (*Figure 7a*). The number of homing gRNAs, which strongly influenced the probability that homing occurred successfully, was also influential (*Figure 7a*). Assumptions regarding the sub-fertility of XO females had less influence on simulation results. Partial dependency plots illustrated that the placement of the Y-shredding drive within a haploinsufficient gene robust to $P_N$ up to 0.1, at least for a starting population size of 10,000 mice, but higher probabilities of NHEJ produced low probabilities of eradication (*Figure 7b*). Interestingly, when single-site mutations arising through NHEJ were not assumed to guarantee non-functionality of the developmental gene (i.e. $P_{nf} < 1$), probabilities of eradication peaked when the Y-shredding efficiency was also less than 1.

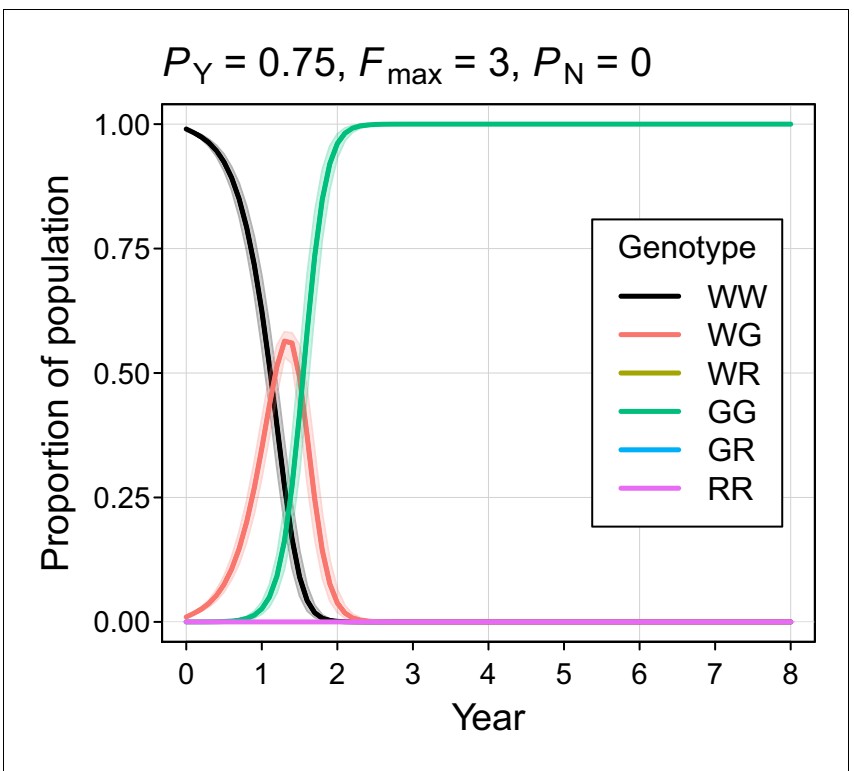

**Figure 4.** Autosome genotype trajectories (mean ±95% confidence intervals) for an example parameterization that resulted in permanent suppression (but not eradication) of the mouse population. Abbreviations denoting different alleles are: W, wildtype; G, gene drive, R, resistant. (Note that resistant alleles cannot be created when $P_N = 0$.).

DOI: https://doi.org/10.7554/eLife.41873.006

## Discussion

The successful eradication of vertebrate pests on islands has been greatly aided by the adoption of new combinatorial technologies (*Howald et al., 2007*; *Gregory et al., 2014*). Yet, new innovative tools are required to complement existing approaches and extend eradication efforts to increasingly large target populations (*Campbell et al., 2015*). The targeted deletion of sex chromosomes through 'programmable' endonucleases is now feasible using the CRISPR genome-editing system (*Adikusuma et al., 2017*; *Zuo et al., 2017*) and, when coupled with a homing mechanism, offers a new option for distorting the sex ratios of pest populations. This Y-CHOPE gene-drive strategy requires two different endonuclease systems to drive the homing and chromosome shredding activities. Ideally, Y-chromosome deletion should occur as early as possible during embryogenesis to ensure that all primordial germ cells lack the Y-chromosome. Here, we show for the first time that the CRISPR/Cas12a system can be utilized to delete the Y chromosome in cultured cells derived from the early murine embryo. Indeed, the Y-shredding efficiency of CRISPR/Cas12a Centro 37X-A gRNA (~90%) was slightly higher than that achieved using the CRISPR/Cas9 system. It is possible that Y chromosome elimination could be further improved by screening additional gRNAs and/or multiplexing. Our data suggest that it may be feasible to generate a Y-CHOPE gene drive using CRISPR/Cas9 for homing and CRISPR/Cas12a for Y shredding. While such a gene drive cassette would be large, it is worth noting that efficient homing of a 17 kb gene drive cassette (*Gantz et al., 2015*) has been demonstrated in mosquitoes, suggesting that size may not be a barrier to efficient homing. That being said, it remains unclear whether efficient homing can be achieved in rodents, with recent data suggesting that significant optimisation will be required (*Grunwald et al., 2018*), even for cassettes of modest proportions.

Our *in silico* results demonstrate that a Y-CHOPE gene drive is capable of eradicating a pest vertebrate population (*Figure 2 and 3*). However, the performance of this drive *in silico* was sensitive to

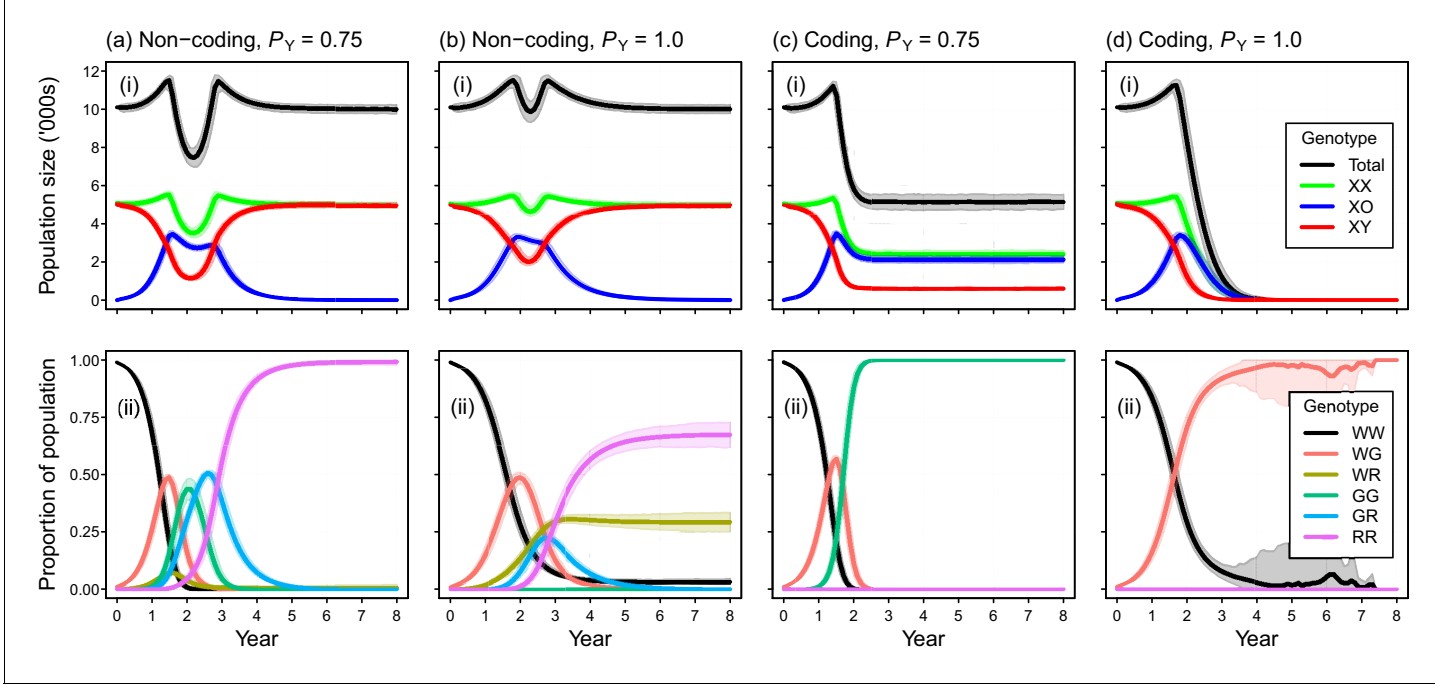

**Figure 5.** Simulated efficacy of Y-chromosome shredding gene drive when resistant alleles are created by non-homologous end-joining during homing. Simulated trajectories (mean ±95% confidence intervals) are shown for two different Y-shredding efficiencies ($P_Y$), assuming the drive was positioned within a non-coding (a, b) or coding region (c, d). Total population and allosome genotype trajectories (i), and the proportion of different autosome genotypes (ii), are shown for each scenario. In the bottom right panel, all simulated populations were eradicated within 8 years. All simulations assumed an initial population size of 10,000 mice, inoculation with 100 gene-drive carrying XX females, polygynous breeding ($F_{max} = 3$), and substantial opportunity for the creation of resistant alleles through non-homologous end-joining ($P_N = 0.1$). In (c,d), the gene drive is assumed to be positioned within a haploinsufficient critical gene, such that resistance alleles cause early embryonic lethality ($P_{nf} = 1$). Abbreviations denoting different autosome alleles are: W, wildtype; G, gene drive, R, resistant.

DOI: https://doi.org/10.7554/eLife.41873.007

the efficiency of the Y-shredding machinery and the mating system of the target species. Population control through a Y-chromosome shredding gene drive relies on limiting the availability of males and therefore reducing the proportion of females that reproduce each breeding cycle. Our simulations indicate that, for polygynous species such as mice, Y-shredding efficiencies must be greater than *c.* 0.9 to produce high probabilities of eradication success (*Figure 6*). The efficient Y-shredding activity of the CRISPR/Cas12a Centro 37X-A gRNA in ES cells suggests that it may be possible to achieve Y-shredding efficiencies of >90% in the germ line *in vivo*, particularly if expression of the endonuclease and gRNA is constitutive. When the Y-shredding efficiency was too low to produce eradication, sex-ratio distortion altered the population's reproductive output and lead to the establishment of a new carrying capacity. Under a low level of polygynous breeding, the carrying capacity of the population was reduced; that is, population suppression was achieved because the mean proportion of females breeding was decreased. Assuming a highly polygynous mating system, however, long-term population increase could result because male limitation was never achieved, and the bias toward females increased the reproductive potential of the population.

The capacity to develop a gene drive that permanently suppresses (but does not eradicate) a target pest population raises interesting questions about the adoption of such technology. Eradication failure in our simulations was not due to the failure of the gene drive to spread. In fact, population suppression (but not eradication) could result despite the drive spreading to fixation if the Y-shredding efficiency was low, because the negative impact of sex-ratio distortion could not overcome the improvement in survival rates at low population sizes which was assumed by our model. Permanent suppression of a pest population accompanied by the indefinite persistence of a gene drive could be considered a high-risk strategy that maximises the potential for the human-mediated transport of gene-drive carriers beyond a target population (*Esvelt et al., 2014*; *Campbell et al., 2015*). On the

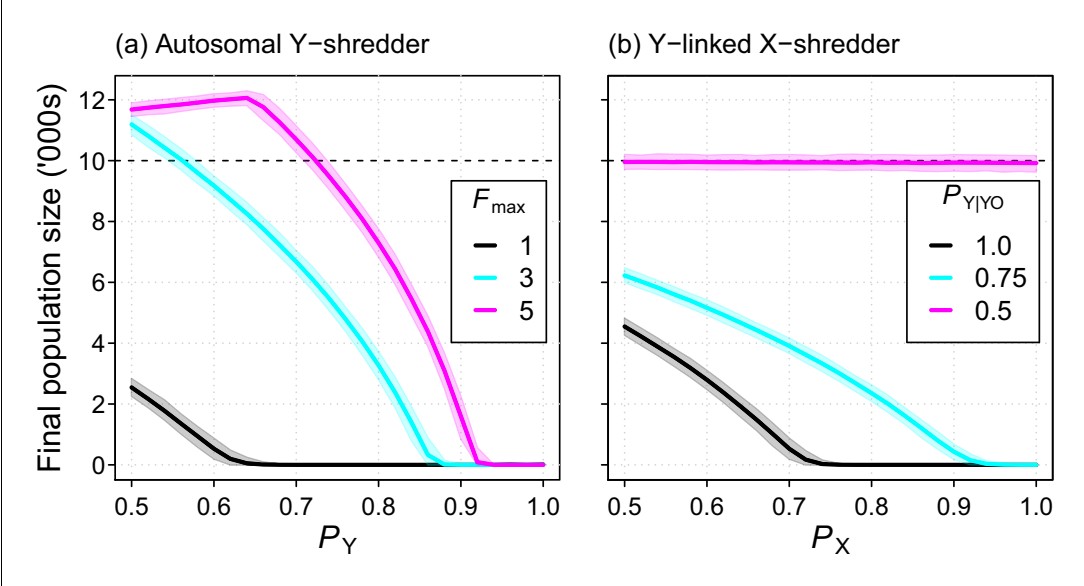

**Figure 6.** Final population size (mean ±95% confidence intervals) after 10 year simulations for two different chromosome-shredding gene-drive strategies. (**a**) An autosomally integrated Y-CHOPE drive. Final population size is shown as a function of the Y-shredding efficiency ($P_Y$) and the level of polygyny assumed for mice ($F_{max}$), assuming placement of the gene drive within the coding region of a haploinsufficient developmental gene, and the baseline parameterisation (see *Table 1*). (**b**) An X-shredding Y-drive. Results are shown as a function of the X-shredding efficiency ($P_X$), and the probability of offspring inheriting a Y-chromosome from male gene-drive carriers that is germline YO ($P_{Y|YO}$). The initial population size of 10,000 mice is indicated (horizontal dashed line).

DOI: https://doi.org/10.7554/eLife.41873.008

other hand, gene-drive persistence following suppression would guard against the migration of wild type individuals into the managed area. Finally, gene-drive-mediated suppression could potentially be used as one part of a larger eradication strategy that included a range of existing control options (e.g. trapping, poison baiting) (*Russell and Richardson, 1971*; *Campbell et al., 2015*). Given that the destruction of gene-drive carrying animals would be undesirable, however, additional simulation modeling is required to investigate the likely impact of combining gene drives with additional control activities.

**Table 1.** Details of the baseline parameterization of the individual-based model, and the parameter ranges tested through sensitivity analysis using Latin hypercube sampling.

| Parameter | Baseline | Sensitivity Analysis |
|---|---|---|
| Probability of Cas9 cutting ($P_C$) | 0.95 | $U(0.7, 1)$ |
| Probability of NHEJ ($P_N$) | 0, 0.1 | $U(0, 0.5)$ |
| Number of guide RNAs (*nGuides*) | 3 | $U(1, 5)*$ |
| Probability of mutation causing non-functionality ($P_{nf}$) | 1 | $U(0.66, 1)$ |
| Y-shredding efficiency ($P_Y$) | 0.5–1 | $U(0.5, 1)$ |
| Fertility multiplier for XO females (*XOfertility*) | 0.6 | $U(0, 1)$ |
| X:O bias in inheritance ($P_{X|XO}$) | 0.66 | $U(0.5, 1)$ |
| Maximum number of female mates per male ($F_{max}$) | 1, 3 | $U(1, 10)*$ |
| Mean litter size (*m*) | 6 | $U(2, 10)$ |
| Maximum annual population growth rate ($r_{max}$) | 7.97 | $U(6, 9)$ |

*For the sensitivity analysis, these parameters were sampled from discrete uniform distributions.

DOI: https://doi.org/10.7554/eLife.41873.011

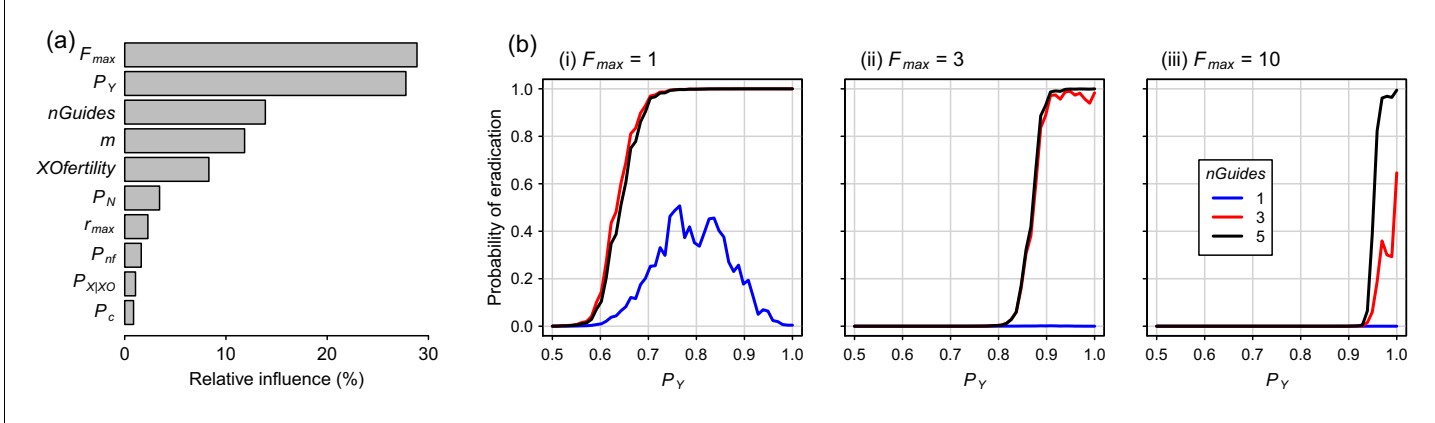

**Figure 7.** Boosted regression tree (BRT) summary of the sensitivity analysis for the Y-CHOPE gene drive. (**a**) Relative influence of the ten simulation parameters included in the sensitivity analysis (parameter abbreviations are provided in *Table 1*). (**b**) Predicted probabilities of eradication derived from the BRT model for three different levels of polygyny ranging from a maximum of one to 10 females mates per male (i-iii). These partial dependency plots assume the following seven less influential parameters were set as follows: $m = 6$, $XOfertility = 0.6$, $P_n = 0.1$, $r_{max} = 7.76$, $P_{nf} = 0.66$, $nGuides = 3$, $P_{X|XO} = 0.66$, and $P_C = 0.95$.

DOI: https://doi.org/10.7554/eLife.41873.009

The following figure supplement is available for figure 7:

**Figure supplement 1.** The cross-validation deviance (**a**) and stability of sensitivity measures (**b**) derived from BRT emulators as the number of parameter samples used for sensitivity analysis is increased.

DOI: https://doi.org/10.7554/eLife.41873.010

Self-replication errors during homing, which arise due to NHEJ-mediated repair, can lead to the creation of resistance alleles that can never acquire the gene drive (*Eckhoff et al., 2017*; *Champer et al., 2018*). Although some empirical research suggests NHEJ occurs rarely (*Akbari et al., 2015*; *DiCarlo et al., 2015*; *Gantz et al., 2015*; *Hammond et al., 2016*), one recent study suggests that the rate of NHEJ-mediated repair could be as high as 40% in the fruit fly *Drosophila melanogaster* (*Champer et al., 2018*). Simulation modeling of gene drives in mosquitoes and mice indicates that this repair pathway threatens the viability of this technology for pest control, if resistant alleles have no (or little) fitness cost (*Eckhoff et al., 2017*; *Prowse et al., 2017*). Our current results show that a Y-chromosome shredding gene drive placed within a non-coding region is similarly vulnerable: after an initial period of sex-ratio distortion, the population is expected to rebound to carrying capacity after the evolution of resistant alleles and selection favouring their spread, even when the Y-shredding efficiency is 100% (*Figure 5a,b*). Positioning the drive within a haploinsufficient gene could solve this problem, at least in theory, if any mutation arising through NHEJ causes lethality early in development, because these mutations will actually help with population suppression (*Figure 5c,d*). In support of this approach, recent experimental data for mosquitoes indicated that the NHEJ-mediated mutation of a functionally constrained target sequence produced nuclease-resistant genes which were non-functional and therefore not favoured by selection (*Kyrou et al., 2018*). However, empirical data are required to demonstrate the feasibility of such a strategy in rodents.

Ablation of entire sex chromosomes using the CRISPR genome-editing system simply requires an additional orthogonal programmable endonuclease and guide RNA(s) that target the allosome (*Galizi et al., 2016*; *Adikusuma et al., 2017*; *Zuo et al., 2017*). Therefore, this approach alleviates the need to incorporate functional sex-reversing genes within the gene-drive cassette, which would require substantial laboratory work-up and testing for rodents (*Prowse et al., 2017*). Although, to date, there has been more interest in X-shredding strategies (*Deredec et al., 2008*; *Galizi et al., 2014*; *Galizi et al., 2016*), we suggest there are potential advantages to a Y-CHOPE drive design. First, an autosomally integrated Y-shredder could be spread by both sexes, whereas a Y-linked X-shredder could only be spread by males. As a result, the population suppression achieved by a Y-shredding design could be superior under certain circumstances, particularly if the pest species targeted was not very polygynous (*Figure 6*). In fact, a Y-chromosome shredding drive is robust to

the sub-fertility of XO females (*Figure 7a*), because the dual-germline homing mechanism allows heterozygotic XX females and XY males to contribute to the drive's spread. From a technical standpoint, an autosomal Y-shredder might be also be simpler to engineer, because the meiotic expression required for an X-shredding drive is complicated by the transcriptional shut down of both sex chromosomes during meiosis (*Galizi et al., 2014*; *Kyrou et al., 2018*). Finally, relative to a male-biasing strategy, a female-biasing approach could be more robust to sexual selection by 'choosy females'. In many vertebrate species, females prefer to choose male mates with certain phenotypic traits, potentially because they confer 'good genes' or attractiveness to their offspring (*Lande, 1981*; *Prokop et al., 2012*). If gene-drive carrying males were to suffer a phenotypic cost, female mating preferences might inhibit the spread of the gene drive spread, but this seems unlikely for a Y-chromosome shredding drive because male limitation should limit female choice.

On the other hand, whereas an X-shredding drive could be driven from the Y chromosome, a Y-shredding strategy requires a functioning homing mechanism. Two further, interrelated disadvantages of a Y-shredding drive are that: (1) population growth could be stimulated initially in polygynous species until males become limiting, because the female-biasing effect of the drive can actually increase the reproductive potential of the population and (2) accurate knowledge of the species' mating system (particularly the number of female mates per male) is required to predict the population-level consequences of a gene-drive release. However, the unwanted impact of initial population growth on island ecosystems could potentially be mitigated by first using traditional control activities to reduce the density of the target species. Interestingly, when $P_{nf}$ <1 such that single-site indels were not guaranteed to cause loss-of-function of the haploinsufficient gene incorporating the drive element, probabilities of eradication could peak for Y-shredding efficiencies less than one when few homing gRNAs were used (*Figure 7b(i)*). This somewhat counterintuitive result reflects the fact that, when $P_Y$ approaches 1, few gene-drive positive XY males are produced. This slows down the spread of the Y-CHOPE drive because homing is essentially occurring in XX and XO females only, which leaves more time for resistance alleles to evolve before eradication is achieved. However, given our maximum empirical estimate of $P_Y$ = 0.9 (current study), this interesting phenomenon poses no barrier to the practical implementation of this technology.

Our simulations assumed a small, panmictic population of mice on an island, and ignore the impact of spatial dynamics on gene-drive spread. In reality, a gene-drive eradication attempt might fail in a spatial metapopulation consisting of multiple subpopulations linked by limited dispersal, because the drive might not reach all subpopulations. Similarly, our models used a simple mating function that assumed males had access to all females in the population, but could only mate with a maximum number of females per breeding cycle. The spread of a gene drive might be limited in wild populations if mice established fixed home ranges and only interacted and mated with local animals. However, on small islands at least, spatially targeted release strategies could potentially ameliorate these spatial effects. Finally, our mating function did not allow multiple paternity (i.e. females mated with one male only per breeding cycle); although unrealistic, this assumption is unlikely to impact model outcomes unless sperm competition was also considered. Although multiple paternity could be simulated in future, this would require additional assumptions regarding the competitive ability of sperm cells carrying a Cas9 seqence and/or that lack a sex chromosome.

One of the primary drawbacks of gene-drive technology is the risk of the dispersal or human-mediated transport of gene-drive carriers beyond the laboratory or the population targeted for management (*Esvelt et al., 2014*). Since gene drives are self-sustaining and theoretically could spread throughout the global distribution of a target species (*Noble et al., 2018*), the development and application of this technology must include appropriate controls (*Esvelt et al., 2014*; *Oye et al., 2014*; *DiCarlo et al., 2015*). To improve the safety of field applications, a number of temporally and/or spatially temporally limited drive strategies have been proposed and modeled, including daisy-chain drives which lose activity over time (*Noble et al., 2016*; *Dhole et al., 2018*) and threshold drives which exploit engineered underdominance to protect non-target populations against gene-drive invasion (*Marshall and Hay, 2012*; *Akbari et al., 2013*; *Dhole et al., 2018*). In theory, our Y-CHOPE drive could be coupled with such self-limiting systems, but substantial empirical research would be required to demonstrate the feasibility of this approach.

The molecular machinery required for the targeted deletion of sex chromosomes is now available (*Galizi et al., 2014*; *Adikusuma et al., 2017*; *Zuo et al., 2017*) and provides a new tool for sex-ratio distortion that could be spread through pest populations by homing endonucleases. Our study

demonstrates that a Y-CHOPE gene drive could be an effective tool for the eradication of alien rodents on islands, and introduces a new approach with both advantages and disadvantages relative to the X-shredding sex-distortion systems that have been proposed previously. However, we also show that the efficiency of Y-shredding needs to be high (e.g. >90% for mice) to guarantee eradication rather than suppression, and that the gene-drive cassette must be positioned carefully within the genome to avoid the evolution of resistant alleles.

## Materials and methods

### Cas12a Y-shredding in mouse embryonic stem cells

A plasmid expressing the *Lachnospiraceae* bacterium (Lb) Cas12a and its gRNA (Plasmid PTE4398 Addgene #74042 (*Tóth et al., 2016*)) was modified to add the T2A-Puromycin resistant cassette at the C terminal of the Cas12a coding sequences to enable puromycin drug resistant selection of transfected cells. In brief, the T2A-Puro fragment was produced by digestion of plasmid PX459.V2 (Addgene #62988 (*Ran et al., 2013*)) using *FseI* and *NotI* (NEB) and the fragment was ligated to the *FseI* and *NotI*-digested PTE4398 backbone to generate a plasmid expressing LbCas12a-T2A-Puro and its gRNA (all-in-one plasmid). Generation of plasmids carrying unique gRNAs was performed using the protocol described by *Tóth et al. (2016)*. In brief, oligonucleotide duplexes containing the guide sequences (*Supplementary file 1*) were ligated into the *BsmBI* golden gate cloning site of the all-in-one plasmid followed by Miraprep plasmid preparation (*Pronobis et al., 2016*) and Sanger sequencing verification. The empty vector (EV) control in this study is the all-in-one plasmid retaining the *BsmBI* golden gate cloning site. gRNAs were identified by manual screening of Y chromosome sequences using the CCTop gRNA design tool (CCTop, RRID:SCR_016890, http://crispr.cos.uni-heidelberg.de; *Stemmer et al., 2015*). gRNA sequences and their Y chromosome locations are provided in *Supplementary file 1*.

Cell culture and plasmid transfection was performed as described previously (*Adikusuma et al., 2017*). Mycoplasma-negative RI mouse embryonic stem cells (RRID:CVCL_2167), sourced from Andras Nagy's laboratory, were used in this study (*Nagy et al., 1993*; *Hughes et al., 2013*). Briefly, R1 mouse ES cells were cultured in 15% FCS/DMEM supplemented with 2 mM Glutamax (Gibco), 100 μM non-essential amino acid (Gibco), 100 μM 2-mercaptoethanol (Sigma), 3 μM CHIR99021 (Sigma) 1 μM PD0325901 (Sigma) and LIF (Millipore). One million ES cells were nucleofected with 3 μg of plasmid DNA using the Neon Transfection System 100 μL Kit (Life technologies) at 1400 V, 10 ms and three pulses. Selection of transfectants was conducted 24 hr later by exposure to puromycin (1 μg/ml, Gibco) for 48 hr. Surviving cells were cultured for 4–7 days without selection before harvesting.

Genomic DNA was extracted using High Pure PCR Template Preparation Kit (Roche) according to the manufacturer's instructions. qPCRs were performed using Fast SYBR Green Master Mix (Applied Biosystems). *Sox1* qPCR was used as internal reference control to normalize qPCR values across samples. To quantify the Y-chromosome dosage, qPCR was performed using primers located at the chromosome termini. All qPCR primers and their positions on the Y chromosome are listed in *Supplementary file 1*.

### Individual-based modeling of a Y-CHOPE gene drive
#### Model overview

We used an individual-based model (IBM) to simulate the capacity of a Y-CHOPE gene drive to accomplish the eradication of a pest mouse (*Mus musculus*) population on an island. The model was constructed in the R software environment for statistical and graphical computing (*R Development Core Team, 2015*). Briefly, we assumed a discrete-time, pre-breeding census design (*Caswell, 2001*) and represented the target mouse population as a collection of individuals characterized by state variables. Within each breeding cycle, individuals were transitioned through the following five stages: (i) mate allocation; (ii) reproduction and inheritance; (iii) Y-shredding and homing of the gene drive within heterozygous individuals; (iv) density-dependent mortality; and (v) aging. The IBM explicitly accounted for both demographic and genetic stochasticity, which become increasingly important as wild animal populations are reduced to a small number of individuals (*Gilpin and Soulé, 1986*). Baseline parameters used to simulate the Y-CHOPE drive are provided in *Table 1*.

## Gene-drive design, homing and inheritance

We assumed a gene-drive cassette located on an autosome, which encoded the orthogonal gRNAs and endonucleases required for homing and for Y-chromosome shredding. Biased inheritance of the gene drive was simulated assuming a dual-germline homing mechanism, such that homing occurred prior to meiosis in the germlines of males and females that were heterozygous for the gene drive. Although HR-mediated repair is required for successful homing, a competing repair pathway termed non-homologous end-joining (NHEJ) can also be used to repair gene-drive-mediated DNA cleavage. NHEJ repair typically generates small insertions or deletions (indels) at the gRNA-binding site. This process can produce 'resistant' alleles that will subsequently resist cleavage by the Cas9 endonuclease, and can never acquire the gene drive. In our model, therefore, the homing rate was assumed to be a function of three parameters: the probability of cutting occurring ($P_C$), the probability of NHEJ conditional on cutting having occurred ($P_N$), and the number of multiplexed homing gRNAs.

Assuming $S$ gRNAs targeting $S$ unique cutting sites, the state into which a wild-type allele can move during germline homing is conditional on the number of susceptible sites it currently possesses. We have previously derived explicit expressions to calculate the probability of a wild-type allele moving from $s$ to $j$ susceptible sites ($P_{sj}$) during homing, under the assumption that Cas9-mediated cutting occurs sequentially (i.e., independently) at each target site (Prowse et al. 2017) or simultaneously, in which case the deletion of long sequence stretches between cutting sites might occur due to NHEJ (Prowse et al. 2018). The probability of successful homing is then given by $1 - \sum_{j=0}^{s} P_{sj}$. In this study, we assumed simultaneous cutting for all simulations, and for each individual we tracked the state of maternal and paternal autosomal alleles which can be considered as: (1) wildtype (W) alleles, with between 1 and S susceptible cutting sites remaining; (2) resistant (R) alleles, with no susceptible sites remaining; or (3) gene-drive alleles. For wildtype and resistant alleles, we also recorded whether the deletion of sequence intervening between two recognition sites had occurred (see below). To simulate germline homing in a heterozygotic parent of genotype WG, for each offspring produced by the model we simulated the modification of the susceptible parental W allele by sampling from multinomial distributions based upon the transition probabilities $P_{sj}$ (*Prowse et al., 2018*). Autosomal alleles and allosomes were then stochastically allocated to offspring from each parent using Bernoulli distributions with $p$ = 0.5 (i.e. Mendelian inheritance), except when an inheritance bias in favour of the X chromosome was assumed for XO female parents (*Table 1*).

We also tested two different placements of the gene-drive cassette. First, we assumed the cassette was placed within a non-coding region. Second, using the strategy proposed by *Esvelt et al. (2014)*, we modeled placement of the gene drive immediately 3' of a haploinsufficient gene. Here, the drive element contains gRNAs that target the 3' coding sequence of the gene as well as a recoded version of the target sequence that retains complete functionality. This positioning should ensure that the majority of individuals that inherit a resistant allele (i.e. mutations arising through NHEJ) will fail to develop, thus preventing the spread of resistance, whilst any animal with the gene drive will be viable. For this latter strategy, we initially assumed a per-mutation probability of gene non-functionality ($P_{nf}$) of 1 (i.e. any single-site mutation formed by NHEJ was sufficient to cause loss-of-function of the target gene), and also that the deletion of intervening sequence between two recognition sites always resulted in gene loss-of-function. Through sensitivity analysis, however, we tested $P_{nf} <1$ such that single-site deletions caused by NHEJ would not necessarily cause loss-of-function of the haploinsufficient gene.

## Y-chromosome shredding

In XY males that inherited at least one gene-drive copy, we assumed shredding of the Y chromosome occurred in the zygote with probability $P_Y$, which represents the efficiency of the Y- shredding machinery *in vivo*. In our model, Y-shredding caused XY males to develop as XO females, which we assumed were sub-fertile. We therefore corrected the expected litter size of XO females with a fertility multiplier equal to 0.6 (*Probst et al., 2008*). Based on empirical data, we assumed the X chromosome was preferentially passed to the offspring of XO mice, with probability $P_{X|XO}$ = 0.66 (*Kaufman, 1972*; *Probst et al., 2008*). YO progeny suffered early embryonic mortality and were therefore never simulated by the model (*Figure 1*).

## Demography and mating system

For simplicity, we assumed equal survival and fertility rates across all age classes, an equal sex ratio at birth, 10 breeding cycles per year and a litter size ($m$) of 6 offspring per female (*Caughley et al., 1998*). To allow for improved survival at low population densities, we modified survival probabilities as a logistic function of population size. The parameters of the logistic function were calculated so that, in the absence of the gene drive and assuming an equal sex ratio, the population was stable ($r = 0$) when at carrying capacity but when reduced to low density achieved the maximum population growth rate estimated for the species ($r_{max} = 7.76$) (*Pech et al., 1999*). To incorporate the effects of demographic stochasticity, we modeled the outcome of all survival probabilities with Bernoulli distributions.

The *Y*-chromosome shredding gene drive relies on male limitation to achieve population suppression, so the efficacy of this strategy will be partly determined by the mating system of the target species. Since mice are polygynous (i.e. males can mate with multiple females), it is possible that population suppression would not result until the sex ratio became strongly biased toward females. To simulate polygynous mating, therefore, we first specified the maximum number of female mates per male $F_{max}$, where $F_{max} = 1$ indicates short-term monogamy and $F_{max} >1$ indicates polygyny. Mating pairs were formed each breeding cycle by randomly sampling a mate for each female from the available pool of males, where this pool was updated recursively to exclude males that had already reached the maximum number of mates allowed. We did not allow multiple paternity (i.e. females mated with one male only per breeding cycle), for simplicity and also because the degree of competition between sperm cells of different genotypes is unknown. Although $F_{max}$ is certainly greater than one for mice, we also tested $F_{max} = 1$ for comparison.

## Baseline scenarios

We initiated each simulated population at a carrying capacity of 10,000 individuals, assuming an equal number of sexually mature males and females. We then simulated the eradication attempt as the addition of 100 females that were somatic heterozygous for the Y-shredding gene drive. To account for demographic and genetic stochasticity, we performed 1000 replicate simulations of the model for each parameterisation tested. For initial scenario testing, we assumed non-coding gene-drive placement, $P_C = 0.95$, $P_N = 0$, and three multiplexed homing gRNAs. We then increased $P_N$ to 0.1 to account for evolution of resistant alleles through NHEJ, and tested positioning of the drive within both non-coding and coding regions.

## Sensitivity analyses

To explore the expected probability of eradication for a wide range of parameters, we conducted a global sensitivity analysis on ten parameters governing gene-drive performance and mouse demography, assuming exonic positioning of the drive (*Table 1*). To ensure adequate coverage of the multi-dimensional parameter space, we used Latin hypercube sampling to generate 100,000 distinct parameter samples, assuming uniform distributions for each parameter (*Table 1*). Latin hypercube sampling is a stratified, space-filling design for generating random samples from a multidimensional distribution, which generalises a Latin-square design to three or more dimensions (*Fang et al., 2006*). In a Latin hypercube, the number of samples equals the number of parameter divisions, meaning we tested 100,000 different values of each continuous parameter.

We then ran a single simulation per parameter sample (*Prowse et al., 2016*), and emulated the sensitivity-analysis output (i.e. the probability of successful simulated eradication) with boosted regression trees (BRT) using functions in the R package dismo (*Hijmans et al., 2013*). BRT can fit complex, non-linear relationships and automatically handle interactions between predictors (*Elith et al., 2008*), so this technique is particularly useful for summarising the output from simulation studies (*Prowse et al., 2013*; *Prowse et al., 2016*). We fit the BRT model with the function gbm.step, using a binomial error and logit link function, a tree complexity of 5, a learning rate of 0.01, and a bag fraction of 0.75 (*Elith et al., 2008*). We calculated relative influence metrics for each input parameter, and used partial dependency plots to examine relationships between the key parameters and the probability of simulating a successful eradication. To confirm that 100,000 parameter samples were sufficient to generate robust sensitivity metrics, we used the approach recommended by *Prowse et al. (2016)*. In brief, we emulated the sensitivity-analysis output for

parameter subsamples of increasing size (i.e. between 1000 and the complete 100,000 samples). We then confirmed that the cross-validatory performance of the emulator and the sensitivity measures derived from it exhibited asymptotic behavior (relative influence metrics for the 10 parameters converged well before 100,000 samples; see *Figure 7—figure supplement 1*).

## Comparison with an X-shredding Y-drive

As a basis for comparison, we also simulated an X-shredding gene-drive strategy, assuming an X-shredding DNA cassette incorporated within the Y chromosome, which was expressed in the germline and, when present, destroyed the X chromosome during meiosis with some probability $P_X$. We also tested different values for the probability of offspring inheriting a Y-chromosome from male gene-drive carriers in which X-shredding has occurred (i.e. from males that are germline YO) ($P_{Y|YO}$). Hence, conditional on X-shredding occurring, with $P_{Y|YO} = 1$ the inheritance of Y-bearing sperm was guaranteed, while with $P_{Y|YO} < 1$ the transmission of O-bearing sperm was possible. All other relevant parameters were the same as the baseline values used for the Y-CHOPE drive, except that each simulated eradication attempt was initiated with the introduction of 100 males carrying the Y-linked drive. We provide the R code for simulating the autosomal Y-shredder and Y-linked X-shredder as a supplementary file to this paper.

## Acknowledgements

This study was funded by a US Defense Advanced Research Projects Agency's grant to PT and PC. We also thank John Godwin for useful discussions that helped improve the manuscript.

## Additional information

### Funding

| Funder | Author |
| --- | --- |
| Defense Advanced Research Projects Agency | Phillip Cassey Paul Thomas |

The funders had no role in study design, data collection and interpretation, or the decision to submit the work for publication.

### Author contributions

Thomas AA Prowse, Conceptualization, Data curation, Software, Formal analysis, Validation, Investigation, Visualization, Methodology, Writing—original draft, Writing—review and editing; Fatwa Adikusuma, Conceptualization, Data curation, Formal analysis, Investigation, Methodology, Writing—review and editing; Phillip Cassey, Conceptualization, Supervision, Funding acquisition, Project administration, Writing—review and editing; Paul Thomas, Conceptualization, Supervision, Funding acquisition, Investigation, Methodology, Project administration, Writing—review and editing; Joshua V Ross, Conceptualization, Resources, Supervision, Investigation, Methodology, Project administration, Writing—review and editing

### Author ORCIDs

Thomas AA Prowse ⬚ http://orcid.org/0000-0002-4093-767X
Fatwa Adikusuma ⬚ http://orcid.org/0000-0003-2163-0514
Paul Thomas ⬚ http://orcid.org/0000-0002-5002-5770
Joshua V Ross ⬚ http://orcid.org/0000-0002-9918-8167

### Decision letter and Author response
Decision letter https://doi.org/10.7554/eLife.41873.016
Author response https://doi.org/10.7554/eLife.41873.017

## Additional files

### Supplementary files

• Source code 1. R code to simulate Y- and X-shredding gene drives in mice.
DOI: https://doi.org/10.7554/eLife.41873.012

• Supplementary file 1. Details of the gRNA sequences (a) and qPCR primers (b) used for the empirical study of Y-shredding efficiency.
DOI: https://doi.org/10.7554/eLife.41873.013

• Transparent reporting form
DOI: https://doi.org/10.7554/eLife.41873.014

### Data availability

The empirical data from our study are provided (as source data for Figure 1).

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
