## [Decision Letter]

Thank you for submitting your article "A Y-chromosome shredding gene drive for controlling pest vertebrate populations" for consideration by *eLife*. Your article has been reviewed by three peer reviewers, and the evaluation has been overseen by a Reviewing Editor and Diethard Tautz as the Senior Editor. The reviewers have opted to remain anonymous.

The reviewers have discussed the reviews with one another and the Reviewing Editor has drafted this decision to help you prepare a revised submission.

Summary:

Prowse and colleagues demonstrate the in vitro efficiency of a CRISPR/Cas12a-mediated "Y chromosome shredder" that is proposed as a means of suppressing invasive rodent populations by reducing the availability of fertile males. The manuscript starts by demonstrating the elimination of Y-linked loci in mouse embryonic stem cells with up to 90% efficiency, and then continues by modeling the population dynamics of a Y-shredder drive under varying parameters in silico. While gene drive strategies using sex ratio distortion systems have been studied and demonstrated previously (e.g. X-shredders in mosquitoes), the approach proposed here of targeting the Y-chromosome is interesting and novel.

Essential revisions:

1) Reference to previous studies:

A main concern with the paper is that it lacks any reference to previous studies of gene drive strategies using sex ratio distortion systems. Such systems have already been studied extensively, and some of them have been demonstrated experimentally (e.g. X-shredders in insects). These omissions make the Y-shredding approach presented in this paper seem more novel than it actually is. The authors should cite some really relevant literature detailing the building of synthetic sex ratio distortion systems in other organisms – in many cases these are also predicated on the specific nuclease-mediated ablation of sex chromosomes (usually the female sex chromosome) through the targeting of repeated sequences. While in this case it is the Y chromosome that is being targeted (with drive – the shredding of Y in mice itself is not novel and was performed previously by some of these authors), the novelty aspect is somewhat diminished in view of the prior art. The authors should revise the relevant section in the manuscript to make this more clear and not overstate the novelty of their approach.

2) Extension and better description of the modeling analysis:

The modeling analysis is clearly the main thrust of this paper, given that the authors do not actually build a functioning drive system. As such, however, this analysis should be made more comprehensive and also needs to be much better described. In its present form, this study is not repeatable.

We specifically urge the authors to compare the performance of their method against X-shredders, which are not expected to suffer from some of the stated disadvantages of this approach (in particular the reliance on males become limiting).

There also seems to be no particular reason to cap F (maximum number of females that a male can mate with) at 5. Either justify it, or make a theoretical estimate based on overlapping home ranges for the mice and also probe higher value such as 10 or more to take into account initially good opportunities to search for mates when males are rare. Better yet, identify the exact level at which the different drives considered are unable to effectively suppress the populations.

In this paper as well as in Prowse et al., 2017, the demographic parameters were fixed. There should be some investigation into the sensitivity of results due to uncertainty in these values as they can vary widely across populations. This is important for reporting uncertainty on the prediction of effectiveness of the gene drive tool.

The sensitivity analyses and the actual parameters in it are somewhat unclear. It appears that Table 1 indicates the upper and lower limits of parameters (e.g. probability of Cas9 cutting, probability of NHEJ). However, it is not clear which values were tested in this range, if these varied as continuous or discrete values, and whether they were varied independently. Please explain in more detail.

The model used for homing and NHEJ is also not particularly clear. For example, were the populations modeled such that the target allele frequency is the sum of uncut + cut, where cut = homing + NHEJ? Is P_N_ represented as the proportion of all target alleles or the proportion of target alleles that were cut? When P_N_>0, is each of three gRNA target sites considered independently, or does P_N_ reflect a resistant locus (all three gRNA sites are mutated)?

The rules of polygynous mating should be described in more detail. Can there be multiple paternity, and if so what are the genetic rules for that process? If not, how might that affect results (as it is a common process in mice it seems important to discuss). Additionally, did the growth rate of systems with F_max_=1 differ from those of F_max_>1 or were r's adjusted to keep them constant across the different mating structures?

The model structure (equations/pseudocode) and code for implementation should be provided in the supplementary information. Currently, the reader has to refer to Prowse et al., 2017, to understand the general structure of the model, but this doesn't include the advancements presented here.

---

## [Author Response]

Essential revisions:1) Reference to previous studies:A main concern with the paper is that it lacks any reference to previous studies of gene drive strategies using sex ratio distortion systems. Such systems have already been studied extensively, and some of them have been demonstrated experimentally (e.g. X-shredders in insects). These omissions make the Y-shredding approach presented in this paper seem more novel than it actually is. The authors should cite some really relevant literature detailing the building of synthetic sex ratio distortion systems in other organisms – in many cases these are also predicated on the specific nuclease-mediated ablation of sex chromosomes (usually the female sex chromosome) through the targeting of repeated sequences. While in this case it is the Y chromosome that is being targeted (with drive – the shredding of Y in mice itself is not novel and was performed previously by some of these authors), the novelty aspect is somewhat diminished in view of the prior art. The authors should revise the relevant section in the manuscript to make this more clear and not overstate the novelty of their approach.

We have now substantially rewritten the Introduction, to include background on some different sex-distorting approaches, with a focus on the X-shredding research that has been conducted in insects. We have also tried to increase the breadth of the references cited.

Further, for comparison we now explicitly model the performance of an X-shredding gene drive (i.e. a Y-drive), under the same baseline parameters that we use for the Y-shredding drive. These new results are presented in Figure 6, and we compare the relative merits of the two approaches in the Discussion.

2) Extension and better description of the modeling analysis:The modeling analysis is clearly the main thrust of this paper, given that the authors do not actually build a functioning drive system. As such, however, this analysis should be made more comprehensive and also needs to be much better described. In its present form, this study is not repeatable.

We have now included more comprehensive details of modelling methods used in the Materials and methods section, with particular emphasis on explaining the stochastic genetic and demographic processes more thoroughly. Further, as supplementary material we have now provided the R code required to simulate the autosomal Y-shredder and Y-linked X-shredder for different input parameters.

We specifically urge the authors to compare the performance of their method against X-shredders, which are not expected to suffer from some of the stated disadvantages of this approach (in particular the reliance on males become limiting).

As detailed above, we now explicitly compare simulation results obtained for Y- and X-shredders, and we also discuss the advantages and disadvantages of each strategy in the Discussion.

There also seems to be no particular reason to cap F (maximum number of females that a male can mate with) at 5. Either justify it, or make a theoretical estimate based on overlapping home ranges for the mice and also probe higher value such as 10 or more to take into account initially good opportunities to search for mates when males are rare. Better yet, identify the exact level at which the different drives considered are unable to effectively suppress the populations.

We acknowledge that capping *F*_max_ at 5 was an arbitrary maximum that represented a substantial degree of polygyny. Testing *F*_max_ in the range 1-5 was sufficient to demonstrate two important points, namely that the more polygynous mice are assumed to be: (1) the stronger the initial population increase following the gene-drive introduction (Figure 3); and (2) the higher the Y-shredding efficiency needs to be for eradication to be achieved.

However, to take this comment into consideration, we have now tested an *F*_max_ of up to 10 in the sensitivity analysis. Please see Figure 7 for these new results.

In this paper as well as in Prowse et al., 2017, the demographic parameters were fixed. There should be some investigation into the sensitivity of results due to uncertainty in these values as they can vary widely across populations. This is important for reporting uncertainty on the prediction of effectiveness of the gene drive tool.

We have completely redone the sensitivity analysis, which now incorporates the two demographic parameters: litter size (*m*) and the maximum annual population growth rate (*r*_max_). Neither of these parameters had a large impact on the probability of achieving simulated population eradication within 10 years. Please note that maximum and minimum survival probabilities used for the density-dependent survival function could not be modified within the sensitivity analysis, because these values were calculated based upon the values of *m* and *r*_max_.

The sensitivity analyses and the actual parameters in it are somewhat unclear. It appears that Table 1 indicates the upper and lower limits of parameters (e.g. probability of Cas9 cutting, probability of NHEJ). However, it is not clear which values were tested in this range, if these varied as continuous or discrete values, and whether they were varied independently. Please explain in more detail.

As previously detailed in the main text and the caption to Table 1, we used latin hypercube sampling for our sensitivity analysis. This is a standard parameter sampling design which is used to guarantee decent coverage of a continuous multidimensional parameter space. However, as this seems to have caused some confusion, we now provide a brief description of latin hypercube sampling in the Materials and methods. We have also clarified that two parameters (nGuides and *F*_max_) were sampled from discrete uniform distributions for the sensitivity analysis (see note in Table 1).

The model used for homing and NHEJ is also not particularly clear. For example, were the populations modeled such that the target allele frequency is the sum of uncut + cut, where cut = homing + NHEJ? Is P_N_ represented as the proportion of all target alleles or the proportion of target alleles that were cut? When P_N_ >0, is each of three gRNA target sites considered independently, or does P_N_ reflect a resistant locus (all three gRNA sites are mutated)?

As noted in the Materials and methods, we defined *P*_N_ as “the probability of NHEJ conditional on cutting having occurred”. We have previously derived explicit expressions to calculate the probability of a wild-type allele moving from s to j susceptible sites (*P*_sj_) during homing, under the assumption that Cas9-mediated cutting occurs *simultaneously* at each recognition site, in which case the deletion of long sequence stretches between cutting sites can occur due to NHEJ (Prowse et al., 2018). As these expressions are reasonably complicated, we have elected not to reproduce them here, but we have tried to clarify how homing is implemented in our model by expanding on this process in the Materials and methods, and on genetic inheritance more generally. We have clarified in the Materials and methods that a resistant allele is an allele which has lost all susceptible cutting sites, due to NHEJ causing the deletion of single sites, or the longer DNA sequences intervening between two sites. Please note that we are not directly modelling allele frequencies; rather, we are tracking the state of maternal/paternal autosomes and allosomes for each individual represented by the model.

The rules of polygynous mating should be described in more detail. Can there be multiple paternity, and if so what are the genetic rules for that process? If not, how might that affect results (as it is a common process in mice it seems important to discuss). Additionally, did the growth rate of systems with F_max_=1 differ from those of F_max_>1 or were r's adjusted to keep them constant across the different mating structures?

We have now clarified how polygynous mating was implemented in the Materials and methods. In short, multiple paternity was not allowed (i.e., females mated with one male only per breeding cycle), which simplified the model considerably because assumptions regarding the degree of competition between sperm cells of different genotypes were not required. We now discuss the likely impact of this assumption in the Discussion. Further, we have clarified that vital rates were set so the population growth rate *r* was zero for all scenarios when the population was at the carrying capacity (K) and at equal sex ratio (i.e., at initiation of the simulations).

The model structure (equations/pseudocode) and code for implementation should be provided in the supplementary information. Currently, the reader has to refer to Prowse et al., 2017, to understand the general structure of the model, but this doesn't include the advancements presented here.

We have now included the R code as supplementary material. This code allows the user to run simulations for both the autosomally integrated Y-shredding and Y-linked X-shredding gene drives.